# Monitoring a Heatsink Temperature Field Using Raman-Based Distributed Temperature Sensor in a Vacuum and −173 °C Environment

**DOI:** 10.3390/s19194186

**Published:** 2019-09-26

**Authors:** Jingchuan Zhang, Peng Wei, Qingbo Liu

**Affiliations:** 1Beijing Institute of Spacecraft Environment Engineering, Beijing 100094, China; jhw101411@hotmail.com; 2School of Instrumentation and Optoelectronic Engineering, Beihang University, Beijing 100191, China; liuqingbo@buaa.edu.cn

**Keywords:** Raman-based distributed temperature sensor, heatsink, attenuation coefficients parameter

## Abstract

A heatsink is a large experimental device which is used to simulate the outer space environment. In this paper, a Raman-based distributed temperature sensor was used for real-time and continuous heatsink temperature monitoring, and a special Raman-based distributed temperature sensing method and system have been proposed. This method takes advantage of three calibration parameters (Δα,
γ,C) to calculate the temperature. These three parameters are related to the attenuation of the optical fiber, the Raman translation, and the difference of optoelectronic conversion, respectively. Optical time domain reflectometry was used to calculate the location. A series of heatsink temperature measurement experiments were performed in a vacuum and −173 °C environment. When the temperature dropped to −100 °C, the parameter Δα was found to vary. A method was proposed to recalculate Δα and modify the traditional Raman fiber temperature equation. The results of the experiments confirmed the validity of this modified Raman fiber temperature equation. Based on this modified equation, the temperature field in the heatsink was calculated. The Raman-based distributed temperature sensor has potential applications in temperature measurement and judging the occurrence of faults in space exploration.

## 1. Introduction

A heatsink is a device for simulating the extremely cold, vacuum environment in outer-space, and is a key piece of equipment in thermal experiments in space exploration research. To simulate the space environment, the heatsink surface is made of copper pipe, and liquid nitrogen is injected into the pipe. Heatsink temperatures can reach −173 °C [1]. Simulation experiments of satellite thermal radiation can be carried out at the same time.

The traditional monitoring method for heatsinks is to use thermocouple sensors [2], which limits the monitoring area. However, many temperature points must be monitored on the heatsink. Therefore, monitoring the temperature field of a heatsink requires many thermocouples. Moreover, bundles of sensor cables will cause electromagnetic interference [3]. Therefore, traditional thermocouples are not suitable for monitoring the temperature field of a heatsink. In this paper, we used a Raman-based distributed temperature sensor system to monitor the heatsink temperature. 

The heatsink used is shown in Figure 1. It was cylindrical and placed horizontally [4]. The outer diameter of the heatsink was around 3.6 m, the length was around 7.3 m, and the inner diameter was around 3.2 m. At the beginning of the experiment, the gas inside the heatsink was pumped out through an evacuation hole. The lowest pressure in the vessel during the experiment reached 0.0002554 Pa, which can be considered a vacuum, with no gas to conduct heat. The liquid nitrogen then started to circulate around the heatsink. After a period of time, the whole space inside the heatsink was under a vacuum and at −173 °C. The pillars sat on the side of heatsink, and the temperature control device was put on the pillars.

Distributed temperature fiber sensors based on Raman scattering have advantages such as light weight, small size, and immunity to electromagnetic interference [5,6,7]. Most importantly, one distributed temperature sensor can provide many temperature monitoring points in the heatsink. By placing a distributed temperature sensor on the heatsink surface, the temperatures at every point along the optical fiber could be measured almost simultaneously. The points were located by optical time domain reflectometry (OTDR). Therefore, distributed temperature fiber sensors based on Raman scattering are suitable for monitoring the temperature field of a heatsink. At present, the main applications of distributed temperature fiber sensors based on Raman scattering are usually power cables [8], tunnel fire warnings [9], gas pipeline monitoring [10], electric generator insulation faults [11], and applications in other fields such as mines and dams [12,13,14,15]. The sensor is rarely used in the vacuum and −173 °C environment created inside the heatsink. Only a kind of fiber Bragg grating sensor has reportedly been used in a heatsink [16,17].

In this paper, an altered distributed temperature fiber sensor method based on Raman scattering and a calibration method have been proposed for monitoring the temperature field of a heatsink. The experiment results proved the validity of this method.

## 2. Principle of the Raman Distributed Temperature Sensing Method

### 2.1. Principle of the Raman Distributed Temperature Sensing Method in a Normal Environment

Raman backscattered light is used to obtain the temperature distribution along a fiber. The relationships between the intensity of the backscattered light and the temperature at the scattering point are as follows:(1)$$I_{as}=I_{0}K_{as}W\upsilon_{as}^{4}{\left[\exp\left(\frac{h\Delta f}{kT}\right)-1\right]}^{-1}exp\left\{\int_{0}^{L}-\left[\alpha_{0}\left(z\right)+\alpha_{as}\left(z\right)\right]dz\right\}$$
(2)$$I_{s}=I_{0}K_{s}W\upsilon_{s}^{4}{\left[1-\exp\left(-\frac{h\Delta f}{kT}\right)\right]}^{-1}exp\left\{\int_{0}^{L}-\left[\alpha_{0}\left(z\right)+\alpha_{s}\left(z\right)\right]dz\right\}$$
where $I_{as}$ and $I_{s}$ are the intensities of the anti-Stokes and Stokes backscattered light, respectively; $I_{0}$ is the intensity of the incident light; $K_{as}$ and $K_{s}$ are the scattering cross-section coefficients of the anti-Stokes and Stokes scattered light, respectively; W is the optical fiber backscattering factor; $\upsilon_{as}$ and $\upsilon_{s}$ are the frequencies of the anti-Stokes and Stokes backscattered light, respectively; h is Planck’s constant; k is Boltzmann’s constant; Δf is the Raman frequency shift; T is the temperature at the scattering point; $\alpha_{0}\left(z\right)$, $\alpha_{as}\left(z\right)$, and $\alpha_{s}\left(z\right)$ are the attenuation coefficient functions of the incident light, anti-Stokes, and Stokes Raman backscattered light, respectively; and L is the distance from the starting point to the scattering point.

Based on Equations (1) and (2), $I_{as}$ and $I_{s}$ are both influenced by temperature. However, in engineering, $I_{as}$ and $I_{s}$ are also influenced by many other factors, such as fluctuation of the laser, fiber melting, and so on. In order to compensate for the instability factors in the engineering, a voltage ratio R between the anti-Stokes and Stokes backscattered light beams generated by photoelectric detectors was used, and is shown in Equation (3):(3)$$R=\frac{G_{as}}{G_{s}}\frac{K_{as}}{K_{s}}{\left(\frac{v_{as}}{v_{s}}\right)}^{4}exp\left[-\left(\alpha_{as}-\alpha_{s}\right)L\right]exp\left(-\frac{h\Delta f}{kT}\right)$$
where $G_{as}$ and $G_{s}$ are the electrical characteristic parameters of these two photoelectric detectors, respectively. The attenuation coefficient functions $\alpha_{as}\left(z\right)$ and $\alpha_{s}\left(z\right)$ can be simplified as constants $\alpha_{as}$ and $\alpha_{s}$ for engineering purposes [18]. 

Equation (3) can be simplified as follows:(4)$$T=\frac{\gamma }{C+L\Delta \alpha -lnR}$$
where $C=ln\left[\frac{G_{as}}{G_{s}}\frac{K_{as}}{K_{s}}{\left(\frac{v_{as}}{v_{s}}\right)}^{4}\right]$ indicates the difference between the Stokes and anti-Stokes photoelectric conversion channels; Δα = $-\left(\alpha_{as}-\alpha_{s}\right)$ is the difference between the attenuation of the Stokes and anti-Stokes scattering light propagating in the optical fiber; and $\gamma =\frac{h\Delta f}{k}$ indicates the energy drift of the incident pulse light and the scattered light. These parameters are suitable for the whole optical fiber [19].

Based on Equation (4), the Raman distributed temperature sensing system was built, as shown in Figure 2.

In Figure 2, the sensing case was used for photoelectric conversion and temperature demodulation. A pulsed laser generated pulsed light (the wavelength was 1550 nm, the peak power was 2 mW, the pulse width was 15 ns, and the pulse frequency was 10 kHz) based on a clock signal controlled by a computer. The pulsed light penetrated an optical fiber (ordinary multimode optical fiber with acrylate coating and tight sheath) through a wavelength division multiplexer (WDM). When the backscattered light produced in the optical fiber returned along the original fiber, the Stokes (1663 nm) and anti-Stokes (1450 nm) beams were separated by the WDM. The photoelectric detectors (avalanche photon diode) converted the intensity of both the Stokes and anti-Stokes beams into electrical signals, and the signals were collected by data acquisition card 1 (DAQ1). The temperature was demodulated on the computer. In Figure 2, the connection of the optical fiber and the WDM is the starting point O, and the end of optical fiber is point B.

The fiber optical sensor used in the experiment is shown in Figure 2. The starting point O was at 0 m. The length of the optical fiber outside the heatsink was approximately 200 m (from point O to point A), and the length of the optical fiber inside the heatsink was approximately 60 m (from point A to point B). There were optical fiber coils at different locations for calibration experiments, as shown in Figure 2.

### 2.2. Principle of the Calibration Method in a Normal Environment

In Equation (4), there are three parameters, Δα,γ, and C, that need to be calibrated. Δα and γ can be pre-calibrated before the experiments, and they will not change in the normal environment. However, C is different. It will change due to the fluctuation of the APD during the operation period, because it contains parameters related to the APD. Therefore, C needs real-time calibration.

In Figure 2, three optical fiber coils were placed into three different water basins. The positions of the three selected optical fiber coils are *L_1_*, *L_2_*, and *L_3_*, and the temperatures of these three basins were *T_1_*, *T_2_*, and *T_3_*, respectively. 

During the pre-calibration, three equations were calculated as shown in Equation (5)
(5)$$\left[\begin{array}{ccc} 1 & -T_{1} & -T_{1}L_{1}\\ 1 & -T_{2} & -T_{2}L_{2}\\ 1 & -T_{3} & -T_{3}L_{3} \end{array}\right]\left[\begin{array}{c} \gamma \\ C\\ \Delta \alpha \end{array}\right]=\left[\begin{array}{c} -T_{1}lnR_{1}\\ -T_{2}lnR_{2}\\ -T_{3}lnR_{3} \end{array}\right]$$
where *lnR_i_* is the logarithmic ratio of the light power for temperature *T_i_* and location *L_i_*, I = 1,2,3. After pre-calibration, Δα and γ can be obtained by Equation (5). 

In Figure 2, the temperature was collected by data acquisition card 2 (DAQ2) for the real-time calibration, and parameter C was calculated in the computer. We placed the optical fiber coils in the real-time calibration case. Recording the position of the calibration fiber coil $L_{c}$ and the logarithmic ratio of the light power $lnR_{c}$, and measuring the temperature of the real-time calibration case $T_{c}$ by thermocouple, the parameter C could be calculated in real time by the formula below:(6)$$C=lnR_{c}-\Delta \alpha L_{c}+\frac{\gamma }{T_{c}}$$

When the three parameters Δα,γ, and C had been obtained, we could calculate the temperature along the optical fiber sensor according to Equation (4).

### 2.3. Principle of Optical Time Domain Reflectometer

OTDR technology was used to calculate the location of the scattering point on the optical fiber. In Figure 2, the clock controlled the pulsed laser to emit a light pulse and started timing simultaneously. The returning scattered light penetrated the WDM and was collected by a photodetector and DAQ1. The location of the scattering point L can be shown as:(7)$$L=\frac{vt}{2}$$
where v is the speed of light in the optical fiber and t is the time gap between the emitted light pulse and the returning scattered light [20]. Although the speeds of the pump light (1550 nm), Stokes (1660 nm), and anti-Stokes (1450 nm) were different when the length of the optical fiber was 260 m in the experiment, the influence of different speed on location can be neglected in the engineering.

## 3. Experiments and Results

### 3.1. Experiments Environment

Based on Figure 2, the sensing case and real-time calibration case were placed outside the heatsink, as shown in Figure 3. Figure 3 shows that the optical fiber was connected to the real-time calibration case, and entered the heatsink through a special optical fiber connector [21]. The cable left the heatsink through a cable connector.

Figure 4 is a diagram of inside of the heatsink. Figure 5a is a picture of the optical fiber on the left side of the heatsink. Figure 5b shows the optical fiber on the right side of heatsink. Figure 5c shows an enlarged picture of the optical fiber coil, which increased the accuracy of the sensing temperature. Figure 5 shows the optical fiber sensor attached on both sides of the surface by Kapton 3M tape. The tape kept the optical fiber sensor in contact with the surface of the heatsink, which was sufficient for temperature field monitoring inside the heatsink. It was not necessary to stick the optical fiber sensor firmly onto the surface of the heatsink.

Figure 4 and Figure 2 show that the length of the optical fiber inside the heatsink was 60 m. Starting from point O, the fiber segment between 220 m and 225 m was attached on the left side of the heatsink, the 225 m to 235 m fiber was coiled inside the temperature control device, and the 235 m to 260 m fiber was attached on the right side of the heatsink.

Figure 4 shows the temperature control device supported by the pillar, and an enlarged picture of the temperature control device is shown in Figure 6. 

Figure 6a is a picture of the temperature control device. An internal diagram of the temperature control device is shown in Figure 6b. Figure 6b shows four heat plates, four thermocouples (T type, measurement range −200–+350 °C, accuracy 0.1 °C), and one optical fiber coil inside the temperature control device. The optical fiber coil was in good thermal contact with the heat plates and thermocouples. The heat plates adjusted the temperature during the experiment. Compared to the thermal energy produced by the four heat plates, the thermal energy from the thermocouples was small enough to neglect.

The average temperatures measured by the four thermocouples were regarded as standard values and compared with the values of the optical fiber coil inside the temperature control device.

### 3.2. Experiments

#### 3.2.1. Calibration Experiments in a Normal Environment:

Based on the principles of the calibration method in a normal environment, the calibration data are listed in Table 1:

Based on Table 1 and Equation (5), we obtained:(8)$$\left[\begin{array}{ccc} 1 & -\left(77+273.15\right) & -\left(77+273.15\right)\times 75\\ 1 & -\left(79.5+273.15\right) & -\left(79.5+273.15\right)\times 188\\ 1 & -\left(38+273.15\right) & -\left(38+273.15\right)\times 190 \end{array}\right]\left[\begin{array}{c} \gamma \\ C\\ \Delta \alpha \end{array}\right]=\left[\begin{array}{c} -\left(77+273.15\right)\times \left(-1.037\right)\\ -\left(79.5+273.15\right)\times \left(-1.013\right)\\ -\left(38+273.15\right)\times \left(-1.221\right) \end{array}\right]$$

The calibration result of Equation (8) was:(9)$$\{\begin{array}{c} \Delta \alpha =0.0001\\ \gamma =550.5587\\ C=0.5268 \end{array}$$

Based on Equations (4) and (6), the temperature equation (10) in a normal environment can be written as:(10)$$\{\begin{array}{c} T=\frac{\gamma }{C+\Delta \alpha L-lnR}\\ C=lnR_{c}-\Delta \alpha L_{c}+\frac{\gamma }{T_{c}} \end{array}$$
where $L_{c}=135m,\Delta \alpha =0.0001,\mathrm{and}\gamma =550.5587.$

#### 3.2.2. Calibration Experiments in the Heatsink

Based on Equation (10), the temperatures measured by the optical fiber and thermocouples in the temperature control device (as shown in Figure 6b) are compared in Figure 7. The blue curve in Figure 7 is the average temperature of the four thermocouples, and the red curve is the average temperature of the optical fiber. Figure 7a shows that the temperature began to decrease in the 20th hour. In the 30th hour, the temperature decreased to −100 °C. Before the 30th hour and after the 70th hour, the temperature was higher than −100 °C, and the data of the optical fiber were consistent with those of the thermocouples. From the 30th hour to the 70th hour, a significant error between the data of the optical fiber and the thermocouples appeared when the temperature fell below −100 °C, as shown in Figure 7a and enlarged in Figure 7b. As shown in Figure 7, the temperature in the control device was varied by heat plates to simulate the satellite thermal radiation.

In Figure 7b, M1 and M4 are the extreme points of optical fiber results, and M2 and M3 are the extreme points of the thermocouple results. Comparing M1 with M2, the temperature difference between the two results was 6 °C, and the time difference between the two results was 6 min. Comparing M3 with M4, the temperature difference between the two results was 2 °C, and the time difference between the two results was 2 h. From this figure, we can see that when the temperatures dropped to −100 °C, the temperature measurement results of the optical fiber had errors compared with the thermocouples. The error also led to a time difference of up to 2 h between the extreme points. Considering that the optical fiber was in a low temperature environment, the main cause of the error was the slight bend loss of optical fiber in low temperatures. 

Figure 8 shows the composition and material of the optical fiber. As shown in Figure 8, the cladding of the optical fiber was doped silica (thermal expansion coefficient was almost 5 × 10^−7^K^−1^), and the coating was acrylate (thermal expansion coefficient was 7.5 × 10^−5^K^−1^). When the environment’s temperature was below −100 °C, the mismatch of the thermal expansion coefficients between the cladding and coating generated stress, and the stress made the optical fiber bend slightly. This bending changed the attenuation of the scattered light. The parameter Δα in Equation (10) only represents the attenuation of scattered light above −100 °C. Therefore, Equation (10) caused a 6 °C error between the thermocouple and optical fiber below −100 °C, as shown in Figure 7. The time difference of 2 hours was the effect of the fiber attenuation and thermal conduction. When the temperature was below −100 °C, Δα needed to be recalculated. Equation (11) is an altered form of Equation (10):(11)$$\begin{array}{c} \{\begin{array}{c} T=\frac{550.5587}{C+0.0001L-lnR}\\ C=lnR_{c}-0.0135+\frac{550.5587}{T_{c}} \end{array},T\geq -100{}^{\circ}C\\ \{\begin{array}{c} T=\frac{550.5587}{C+\Delta \alpha_{1}L-lnR}\\ C=lnR_{c}-\Delta \alpha \times 135+\frac{550.5587}{T_{c}} \end{array},T<-100{}^{\circ}C \end{array}$$
where $\Delta \alpha_{1}$ is the attenuation of scattered light when the temperature is lower than −100 °C.

According to Equation (4), lnR is linear with location L:(12)$$lnR=\Delta \alpha L+\left(C-\frac{\gamma }{T}\right)$$

By differentiating Equation (12), we can obtain Equation (13):(13)$$\frac{dlnR}{dL}=\Delta \alpha$$

The blue dots in Figure 9 represent *lnR* at every meter in the heatsink when the temperature was −60 °C, −110 °C, and −120 °C, respectively. The red line represents a straight line calculated by least squares fitting. The slope of fitting line and the mean squared error (MSE) of fitting line are shown in Figure 9. The MSE was calculated by Equation (14):(14)$$\mathrm{MSE}=\frac{1}{m}\overset{m}{\underset{i=1}{\sum }}{\left(y_{i}-\hat{y_{i}}\right)}^{2},i=1,2,\ldots ,m$$
where m is the number of the measurement results, $y_{i}$ is the actual measurement result, and $\hat{y_{i}}$ is the measurement result calculated by fitting [22].

From Figure 9a, we can see that the slope of the curve at −60 °C (d(lnR)/d(L)=0.0001) was the same as the parameter Δα calibrated in the normal environment. Using the same method, we calibrated the slope of the curve at −110 °C and −120 °C, as shown in Figure 9b. In Figure 9a–c, the measurement data had a wave, but the slope of the fitting line was obvious, and the MSE was lower to guarantee the accuracy of the results. Therefore, we calibrated the slopes at different temperatures inside the heatsink, which are shown in Table 2.

From Table 2 we can see that the slope of curve below −100 °C was obviously different from that above −100 °C, which means Δα changed in the harsh environment. We can also see from Figure 7 that significant errors between the results of the optical fiber and the thermocouples appeared when temperature was below −100 °C. Therefore, −100 °C can be seen as a threshold for different attenuation factors Δα, and based on Table 2, Equation (15) can be shown as:(15)$$\begin{array}{c} \{\begin{array}{c} \Delta \alpha \approx 0.0001,T\geq -100{}^{\circ}C\\ \Delta \alpha_{1}\approx 0.000085,T<-100{}^{\circ}C \end{array} \end{array}$$

In Equation (15), Δα is the attenuation above −100 °C and $\Delta \alpha_{1}$ is the attenuation below −100 °C. These new parameters were suitable for the whole optical fiber on the heatsink. 

Based on Equations (11) and (15), the temperature equation can be written as:(16)$$\begin{array}{c} \{\begin{array}{c} T=\frac{550.5587}{C+0.0001L-lnR}\\ C=lnR_{c}-0.0135+\frac{550.5587}{T_{c}} \end{array},T\geq -100{}^{\circ}C\\ \{\begin{array}{c} T=\frac{550.5587}{C+0.000085L-lnR}\\ C=lnR_{c}-0.0135+\frac{550.5587}{T_{c}} \end{array},T<-100{}^{\circ}C \end{array}$$

For the same optical power and same position, the temperature difference calculated by Equations (16) and (10) reached approximately 2 °C. Therefore, variation in the parameters can be seen as the main cause of the errors in Figure 7.

Based on Equation (16), the optical fiber temperature was recalculated as shown in Figure 10. Comparing Figure 7 and Figure 10, it can be seen that from the 30th hour to the 70th hour there was only a small error between the temperature measurement results of the optical fiber and the thermocouple.

Combining Figure 7 and Figure 10, the optical fiber results below −100 °C based on Equations (10) and (16) were compared with the thermocouple results and are shown in Figure 11. In Figure 11, the blue curve is the temperature of the thermocouple, and the red and black curves are the temperature of the optical fiber based on Equations (10) and (16), respectively. From Figure 11 we can see that the results based on Equation (16) were much closer to the standard thermocouple results. This shows that Equation (16) effectively reduced errors that appeared due to various reasons. Although low temperatures could reduce the backscattered light power as low as the noise level, it can be seen from Figure 11 that the error was be reduced by Equation (16).

Some of the errors between the temperature measurement results of the thermocouple and the optical fiber based on Equations (10) and (16) are shown in Table 3. In Table 3, $T_{4}$ is the thermocouple data, $T_{5}$ is the optical fiber data based on Equation (10), and $T_{6}$ is the optical fiber data based on Equation (16). $\Delta E_{1}$ is the error between $T_{4}$ and $T_{6}$. $\Delta E_{2}$ is the error between $T_{4}$ and $T_{5}$. As shown in Table 3, the temperature error was reduced based on Equation (16). 

Based on Table 3 and Figure 11, the error was largely reduced by Equation (16). The error may have been due to the different loss of Stokes and anti-Stokes light in low temperature conditions, reduction of the Raman traces, slightly differently guided fibers, or fibers of another manufacturer, and so on. In our engineering, we just recalibrated Δα in Equation (16), and the experiment results proved the method’s validity.

The temperatures along the optical fiber at certain times are shown in Figure 12. The figure shows that when the optical fiber passed through the connector at approximately 200 m, the temperature decreased dramatically. The temperature of the optical fiber in the temperature control device (225–235 m) was approximately −135 °C. The temperatures of the optical fiber on the left (200–225 m) and right (235–260 m) sides of the heatsink were lower than those in the temperature control device. 

A simulation experiment of satellite thermal radiation was carried out at the same time. In Figure 12, the measurement results of the optical fiber sensor show that the internal temperature of the heatsink was −173 °C, which was higher than the temperature −196 °C of liquid nitrogen. The reason for this lies in the satellite thermal radiation experiment. Thermal radiation is reflected inside the heatsink, which meant that the temperatures of some parts of the fiber on the heatsink were only −173 °C at that time. This means that the temperature field inside the heatsink was influenced by the satellite thermal radiation and liquid nitrogen.

One advantages of the Raman distributed temperature sensing system is the distributed temperature measurement. Therefore, we were able to measure dozens point temperatures using the distributed Raman fiber sensor, and the temperature field of the heatsink was calculated by a single optical fiber. Additionally, we further obtained the distribution of two-dimensional temperature field of heatsink by interpolation method, so as to monitor the working conditions of heatsink in the experiment and judge the occurrence of faults. This was very important to this experiment and to future applications in outer-space aircraft.

## 4. Conclusions

In this paper, a Raman distributed temperature sensing system was designed, and a heatsink temperature field experiment in a low-temperature environment was carried out. The temperature measurement equation was changed, and a new calibration method was proposed for the environment inside the heatsink. The experimental results verified the effectiveness of the Raman distributed temperature sensing system and the method. In the future, we believe that this technology will have potential applications in space exploration.

## Figures and Tables

**Figure 1 sensors-19-04186-f001:**
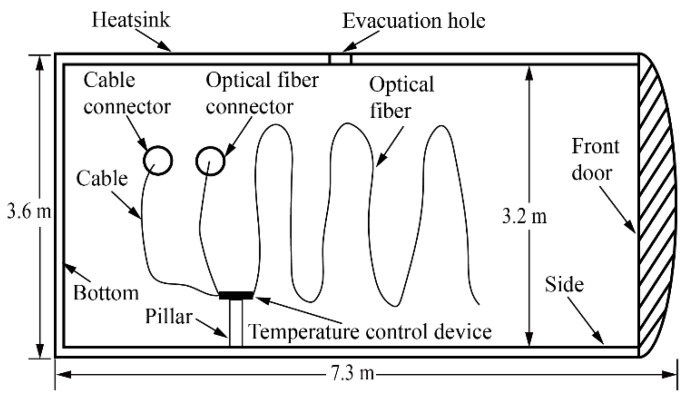
The heatsink.

**Figure 2 sensors-19-04186-f002:**
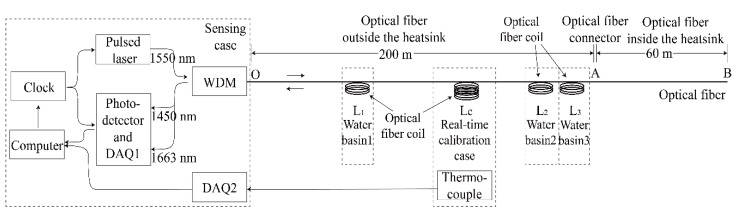
The Raman distributed temperature sensing system.

**Figure 3 sensors-19-04186-f003:**
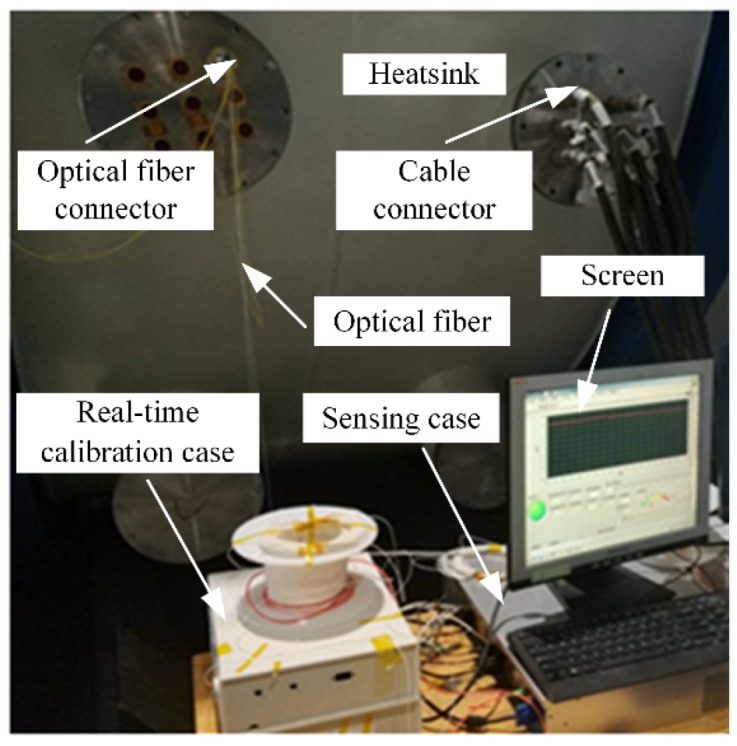
The equipment of the distributed optical fiber Raman temperature sensing system.

**Figure 4 sensors-19-04186-f004:**
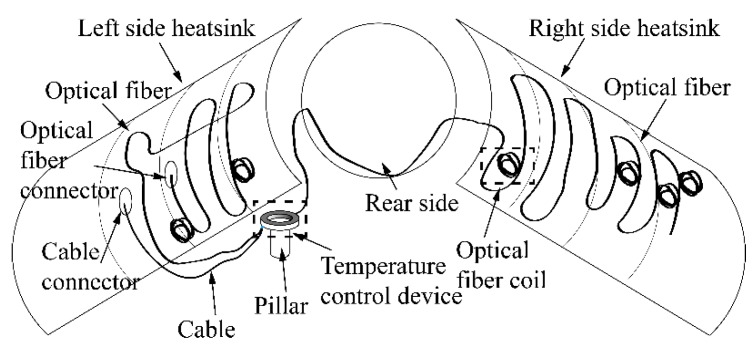
Diagram of the inside of the heatsink.

**Figure 5 sensors-19-04186-f005:**
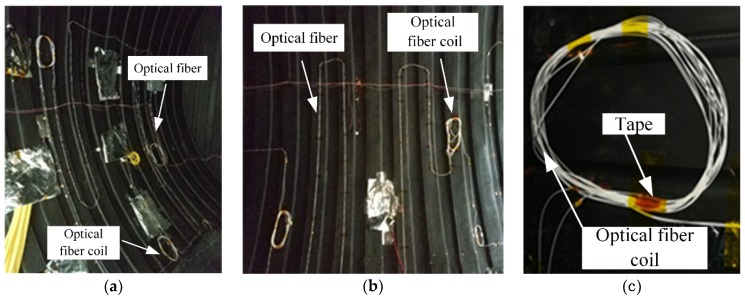
(**a**) Optical fiber on the left side of the heatsink; (**b**) optical fiber on the right side of the heatsink; (**c**) enlarged picture of the optical fiber coil inside the heatsink.

**Figure 6 sensors-19-04186-f006:**
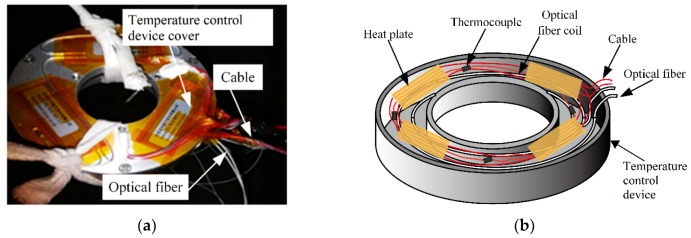
(**a**) Picture of the temperature control device; (**b**) the inside of the temperature control device.

**Figure 7 sensors-19-04186-f007:**
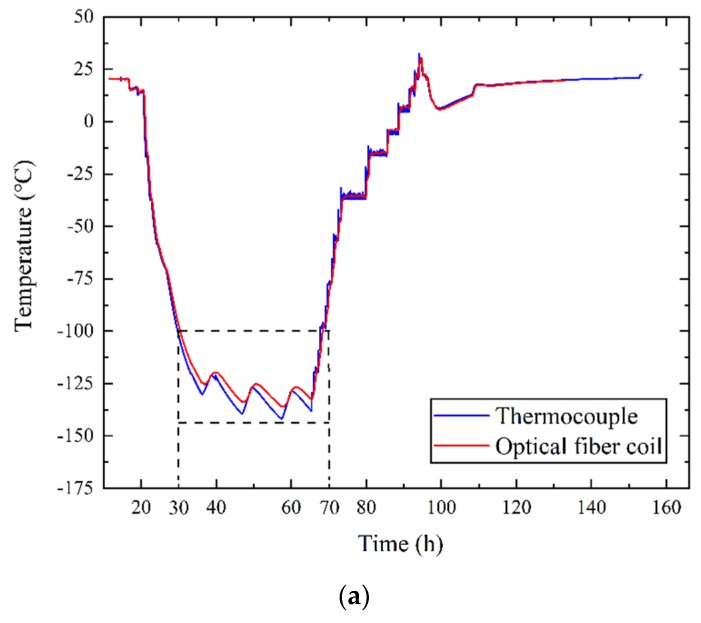
(**a**) The temperatures measured by the optical fiber sensor and thermocouple; (**b**) enlargement of the temperature curve below −100 °C.

**Figure 8 sensors-19-04186-f008:**
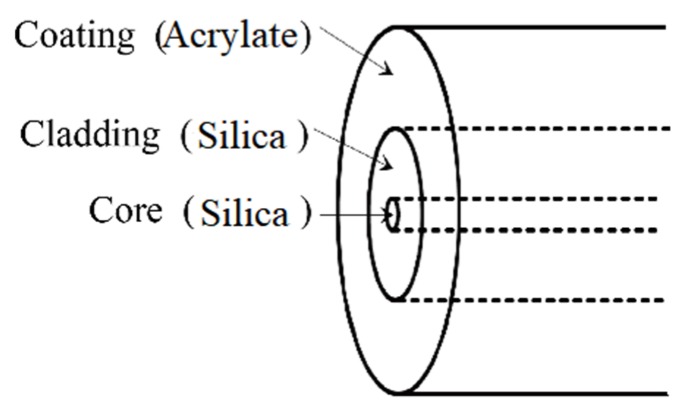
The composition and material of the optical fiber.

**Figure 9 sensors-19-04186-f009:**
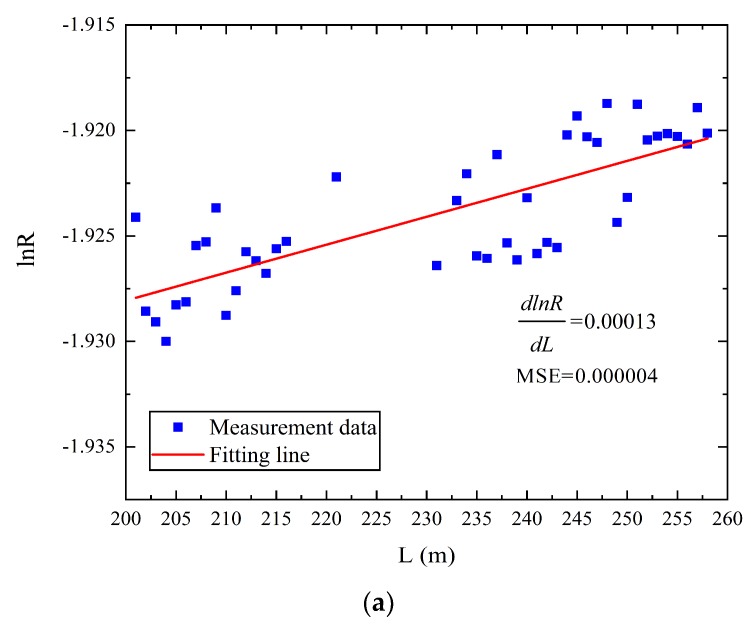
(**a**) Relationship between the location and logarithmic ratio of the optical power at −60 °C;(**b**) relationship between the location and logarithmic ratio of the optical power at −110 °C;(**c**) relationship between the location and logarithmic ratio of the optical power at −120 °C.

**Figure 10 sensors-19-04186-f010:**
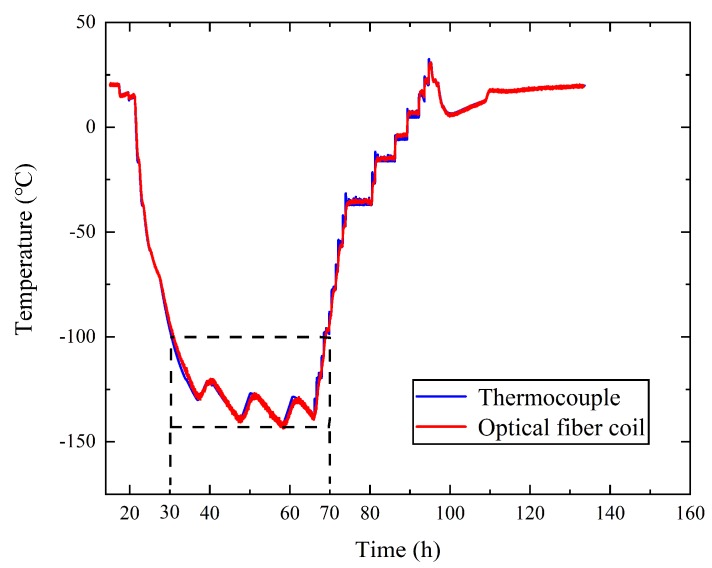
The temperatures measured by the optical fiber sensor and thermocouple after the correction.

**Figure 11 sensors-19-04186-f011:**
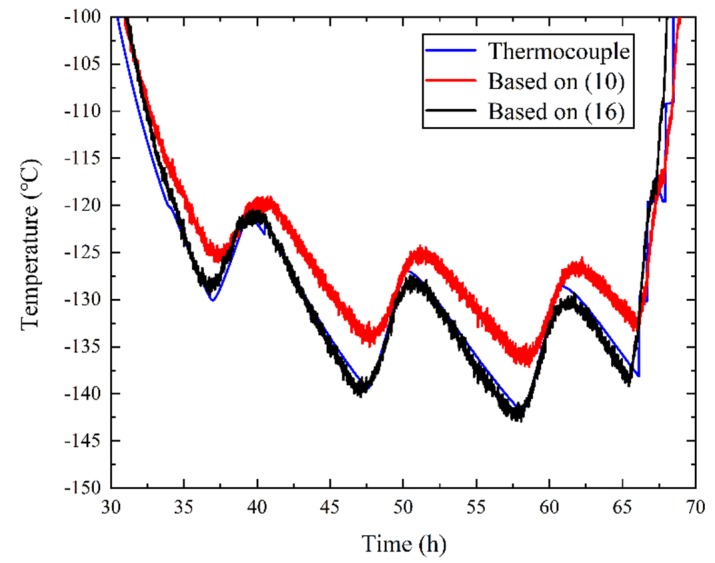
The temperatures measured by the thermocouple and the optical fiber based on Equations (10) and (16).

**Figure 12 sensors-19-04186-f012:**
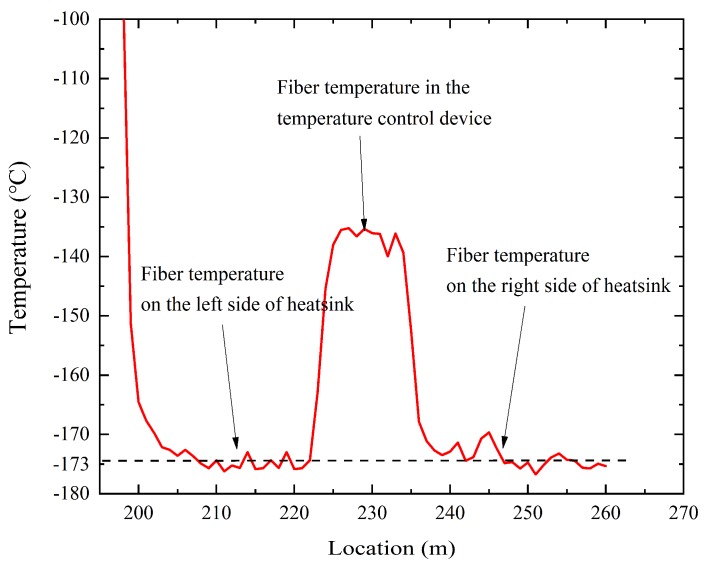
The temperatures along the optical fiber at given times.

**Table 1 sensors-19-04186-t001:** The calibration experiment data.

Temperature (°C)	Location (m)	Logarithmic Ratio of the Optical Power
$T_{1}$ = 77	$L_{1}$ = 75	$lnR_{1}$ = −1.037
$T_{2}=79.5$	$L_{2}$ = 188	$lnR_{2}$ = −1.013
$T_{3}=38$	$L_{3}$ = 190	$lnR_{3}$ = −1.221

**Table 2 sensors-19-04186-t002:** Different slopes at different temperatures in the heatsink.

Temperature (°C)	d(lnR)/d(L)
20	0.00010
−40	0.00013
−60	0.00011
−110	0.000086
−120	0.000085

**Table 3 sensors-19-04186-t003:** Errors based on Equations (10) and (16).

$T_{4}$ (°C)	$T_{5}$ (°C)	$T_{6}$ (°C)	$\Delta E_{1}=|E_{4}-E_{6}|$ (°C)	$\Delta E_{2}=|E_{4}-E_{5}|$ (°C)
−139.326	−134.303	−140.111	0.785	5.023
−137.285	−132.861	−137.984	0.699	4.424
−135.748	−130.656	−135.759	0.011	5.092
−133.608	−128.447	−132.992	0.616	5.161
−128.265	−125.756	−128.867	0.602	2.509
−127.531	−123.203	−127.466	0.065	4.328
−123.758	−120.356	−123.012	0.746	3.402
−118.912	−118.208	−119.587	0.675	0.704

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
