# Peer review of "Monitoring a Heatsink Temperature Field Using Raman-Based Distributed Temperature Sensor in a Vacuum and −173 °C Environment"

_sensors, 2019, doi:10.3390/s19194186_

Round 1

Reviewer 1 Report

The paper "Monitoring the Temperature Field of a Heatsink Using a Distributed Raman Fiber Sensor in a Vacuum and -173°C Environment" describes temperature measurement of a heatsink by DTS system. Based on measurements authors confirmed the suitability of this method for space environment. Paper is well organized into sections. Quality of language and paper readability is good however there are some typos and grammar issues (see comments below). 

Comments - minor issues:

Page 2, line 48 and 57: "Distributed Raman fiber sensors have..." - I would suggest use "Distributed temperature fiber sensors based on the Raman scattering"...or "Raman distributed temperature sensing". Page 4, line 119: "the light powerln" add space after power. Page 6, lines 161 and 162 are identical with lines 164 and 165.  Page 6, table 1: °C seems to be non bold font

Comments - major issues: 

Page 3, line 90: It is not clear if the pulse power 2mW is average or peak power. Fig. 7: the delay between response measured by fiber and by thermocouple seems to be higher than 0.1 hour. Especially in the detail in Figure 2b a difference is 2 hours!   You should mention what is the thermocouple measurement range and accuracy if it is a reference measurement here.

Have you tried modify optical fiber placement? If it is caused by different thermal expansion coefficients then for slightly differently guided fibers or fibers of another manufacturer the new formula may not correspond.

Author Response

To reviewer 1:

Comments - minor issues:

Page 2, line 48 and 57: "Distributed Raman fiber sensors have..."

- I would suggest use "Distributed temperature fiber sensors based on the Raman scattering"...or "Raman distributed temperature sensing".

Answer:

Page 1, line 3, line 15, line 16, line 28 and line 42; Page 2, line 52, line 54, line 55, line 57, line 58, line 59, line 64, line67 and line 68; Page 3, line 90 and line 92; Page 12, line 281; Page 13, line 288 and line 292:

The “Distributed Raman fiber sensors” is replaced as “Distributed temperature fiber sensors based on the Raman scattering” or "Raman distributed temperature sensing" according to the suggestion.

Page 4, line 119: "the light powerln" add space after power.

Answer:

Page 4, line 123: The space is added between power and lnR.

Page 6, lines 161 and 162 are identical with lines 164 and 165. 

Answer:

Page 6, lines 164 and 165 are deleted.

Page 6, table 1: °C seems to be non-bold font.

Answer:

Page 6, table 1: The °C is changed to bold font.

Comments - major issues:  

Page 3, line 90: It is not clear if the pulse power 2mW is average or peak power.

Answer:

Page 3, line 94 is revised according to the suggestion:

“the peak power is 2 mW”.

Fig. 7: the delay between responses measured by fiber and by thermocouple seems to be higher than 0.1 hour. Especially in the detail in Figure 2b a difference is 2 hours!  You should mention what is the thermocouple measurement range and accuracy if it is a reference measurement here. Have you tried modify optical fiber placement? If it is caused by different thermal expansion coefficients then for slightly differently guided fibers or fibers of another manufacturer the new formula may not correspond.

Answer:

Page 7, line 191 is revised according to the suggestion:

“In Figure 7(b), M1 and M4 are the extreme points of optical fiber results, and M2 and M3 are the extreme points of thermocouple results. Comparing M1 with M2, the temperature difference between the two results is 6°C, and the time difference between the two results is 6 minutes. Comparing M3 with M4, the temperature difference between the two results is 2°C, and the time difference between the two results is 2 hours. From this figure we can see that when the temperatures drop to -100°C, the temperature measurement results of optical fiber have errors compared with thermocouple. The error also leads to the time difference up to 2 hours between the extreme points. Considering that the optical fiber is in low temperature environment, the main cause of the error is the slight bend loss of optical fiber in low temperature. ”

Page 5, line 161 is revised according to the suggestion:

“four thermocouples (T type, measurement range -200~+350°C, accuracy 0.1°C).”

Page 12, line 262 is revised according to the suggestion:

“Based on the Table 3 and Figure 11, the error is largely reduced by equation (16). The error may be due to the different loss of the Stokes and anti-Stokes light in the low temperature condition, or reduction of the Raman traces, or slightly differently guided fibers, or fibers of another manufacturer and so on. But in the engineering we can just recalibrate Δα in equation (16). And the experiment results prove the method validity.”

Reviewer 2 Report

The paper is related to application of distributed Raman fibre sensor for measurement temperature of heatsink in space. The paper is very interesting and well written.

Remarks:

1) Abstract. The calibration parameters are given by symbols only. There should be explanation of the symbols, explaining what parameters are hidden under the symbols.

2) Fig. 2. Instead ‘heatink’ should be ‘heatsink’.

3) Eq. 5. It will be better to explain lnR1, lnR2 lnR3 as ‘lnRi is the logarithmic ratio of the light power for temperature Ti and location Li, i=1,2,3’. It allows explaining all parameters.

4) Eq. 5. lnR2 and lnR3 are the same as lnR1. Is it related to their values only or whole definition (that means L2=L3=L1 etc.)?

5) Page 5 line142-143. Too many ‘firmly’ in one sentence.

Author Response

To reviewer 2:

The calibration parameters are given by symbols only. There should be explanation of the symbols, explaining what parameters are hidden under the symbols.

Answer:

Page 1, lines 18 to 20 are added according to the suggestion:

“These three parameters are related to the attenuation of optical fiber, the Raman translation and the difference of optoelectronic conversion, respectively.”

2. Instead “heatink” should be “heatsink”.

Answer:

Page 3, Figure 2 is revised according to the suggestion:

The “heatink” in Figure 2 is revised as “heatsink”.

5. It will be better to explain lnR1, lnR2 lnR3 as ‘lnRi is the logarithmic ratio of the light power for temperature Ti and location Li, i=1,2,3’. It allows explaining all parameters.

Answer:

Page 4, line 118 is revised according to the suggestion:

“where lnRis the logarithmic ratio of the light power for temperature Ti and location Li, i=1,2,3.”

5. lnR2 and lnR3 are the same as lnR1. Is it related to their values only or whole definition (that means L2=L3=L1 etc.)?

Answer:

Page 4, line 118 is revised according to the suggestion:

“where lnRis the logarithmic ratio of the light power for temperature Ti and location Li, i=1,2,3.”

Page 5 line142-143. Too many ‘firmly’ in one sentence.

Answer:

Page 5, line 148 is revised according to the suggestion:

“It is not necessary to stick the optical fiber sensor firmly on the surface of the heatsink.”

Author Response

To reviewer 3:

Terminology: The term heatsink makes me confused at the beginning. Generally speaking, it is referred to a passive thermal exchanger in electronic devices. So I don’t understand that why the heatsink is “a key piece of equipment in space exploration” and “must be monitored in vacuum and -173 degree environment.” A clear definition of a heatsink, as well as its importance and applications, should be introduced firstly. Moreover, in sensing field OTDR is usually referred to “optical time domain reflectometry/reflectometer”, instead of “reflection”, while the “distributed Raman fiber sensor” is usually called “Raman-based distributed temperature sensor/sensing (Raman DTS)”. I encourage the authors using the common terminology in sensing field, since the paper is submitted to the journal

Answer:

Page 1, line 14 is revised according to the suggestion:

“The heatsink is a large experiment device which is used to simulate the outer space environment.”

Page 1, lines 32 to 35 are revised according to the suggestion:

“The heatsink is a device for simulating cold and dark environment in space, which is a key equipment in the vacuum thermal experiment space exploration. To simulate the space environment, the heatsink surface is made by copper pipe, and the liquid nitrogen is injected in the pipe. The heatsink temperature can reach to -173°C environment”

Page 1, line 3, line 15, line 16, line 28 and line 42; Page 2, line 52, line 54, line 55, line 57, line 58, line 59, line 64, line67 and line 68; Page 3, line 90 and line 92; Page 12, line 281; Page 13, line 288 and line 292:

The “Distributed Raman fiber sensors” is replaced as “Distributed temperature fiber sensors based on the Raman scattering” or "Raman distributed temperature sensing" according to the suggestion.

Page 1, line 20, Page 2, line 57 and Page 4, line 127:

The phrase “optical time domain reflection” is revised as “optical time domain reflectometer” according to the suggestion.

Motivation: It is known that the Raman DTS is advantageous in monitoring lots of temperature points. But I doubt that why a heatsink needs so many measurement points. A heatsink, according to the paper, is used for thermal vacuum tests of satellites. It is cooled down by the liquid nitrogen and is vacuum inside, so its temperature should be affected by the liquid nitrogen and thermal radiation only. It is not a very complex scenario and seems that several thermocouples can handle it. The authors should give a convincing reason for the motivation of this paper.

Answer:

Page 1, line 26 is revised according to the suggestion:

“The Raman-based distributed temperature sensor has potential temperature measurement and judge the occurrence of faults applications in space exploration.”

Page 12, line 284 is revised according to the suggestion:

“Besides, we can further obtain the distribution of two-dimensional temperature field of heatsink by interpolation method, so as to monitor the working condition of heatsink in the experiment and judge the occurrence of faults. It's very important to the experiment and the outer-space aircraft.”

Analysis: The authors observe that below -100 degree there was a large error between DTS and thermocouple. However, I would like to see the Stokes and anti-Stokes light in the low temperature condition, which can tell which part (Stokes or anti-Stokes) suffer from extra loss (or both). Meanwhile, low-temperature sometimes may reduce the Raman traces as low as the noise level. This may also contribute the error below -100 degree.

Another issue is the explanation of this phenomenon. The author claim that it is due to the extra stress under low temperature. Does this happen for a straight fiber segment? Does this experiment repeatable? Does the re-calibrated  remains effective for different coil diameters? These additional analysis should be addressed to make the paper clear.

Answer:

Page 12, line 262 is revised according to the suggestion:

“Based on the Table 3 and Figure 11, the error is largely reduced by equation (16). The error may be due to the different loss of the Stokes and anti-Stokes light in the low temperature condition, or reduction of the Raman traces, or slightly differently guided fibers, or fibers of another manufacturer and so on. But in the engineering we can just recalibrate Δα in equation (16). And the experiment results prove the method validity.”

Line 14, “must to be monitored”

Answer:

Page 1, line 14 is revised according to the suggestion:

“The heatsink will be in a vacuum and -173°C environment during the vacuum thermal experiment.”

Line 14, the whole sentence read very weird. Why the heatsink must be monitored in vacuum and -173 degree environment?

Answer:

Page 1, line 14 is revised according to the suggestion:

“The heatsink is a large experiment device which is used to simulate the outer space environment.”

Line 19, “is found to change varied”.

Answer:

Page 1, line 22 is revised according to the suggestion:

“is found varied”.

Line 22 to 23: reads weird, please revise.

Answer:

Page 1, line 24 is revised according to the suggestion:

“The results of the experiments confirm the validity of this modified Raman fiber temperature equation. Based on this modified equation, the temperature field in the heatsink is calculated.”

Line 30: “use thermocouple sensors [2], and while the thermocouple “

Answer:

Page 1, line 37 is revised according to the suggestion:

“while the monitoring area is limited.”

Line 44: “vacuum state condition” vacuum state is a term in quantum physics. Try to use vacuum condition to avoid ambiguity.

Answer:

Page 1, line 48 is revised according to the suggestion:

“which can be considered as a vacuum condition”.

Line 67 to 68: equations numbers: left parentheses missing

Answer:

The left parentheses of equations numbers are added.

Equation (1) - (3), not rigorous. In most cases there are many additional losses in the fiber, such as bending, splice points, etc. So the attenuation should be expressed as .

Answer:

Page 2, Equation (1) - (2) is revised according to the suggestion:

Page 3, line 83 is revised according to the suggestion:

“The attenuation coefficient functions α_as (z) and α_s (z) can be simplified as constant α_as and α_s in the engineering.”

Line 95: what kind of PD is used in the experiment? A normal PD? or a APD, or a SPD?

Answer:

Page 3, line 99 is revised according to the suggestion:

“The APD (avalanche photon diode)”.

Line 73 and 80: duplicate definition

Answer:

Page 3, line 82 is deleted according to the suggestion.

Line 77: “R of between the anti-Stokes”.

Answer:

Page 3, line 80 is revised according to the suggestion:

“the voltage ratio R between the anti-Stokes and Stokes backscattered light”.

Line 90: “based on a clock signal, and the clock is controlled”.

Answer:

Page 3, line 95 is revised according to the suggestion:

“based on a clock signal controlled by a computer”.

Line 107: why the fluctuation of laser will change C? If the laser fluctuates, it will affect the power of both Stokes and anti-Stokes light, but their ratio may keep constant. Please give details.

Answer:

Page 3, line 112 is revised according to the suggestion:

“It will change due to the fluctuation of the APD during the operation period because it contains parameters related to the APD.”

Line 114: “lnR2 and lnR3 are the same defined in the same way as lnR1”.

Answer:

Page 4, line 118 is revised according to the suggestion:

“where lnRis the logarithmic ratio of the light power for temperature Ti and location Li, i=1,2,3.”

Equation (7), not rigorous, since the speed for pump light (1550nm), Stokes (1660nm), and
anti-Stokes (1450nm) are different. Please revise this equation.

Answer:

Page 4, line 133 is revised according to the suggestion:

“Although the speed of pump light (1550nm), Stokes (1660nm), and anti-Stokes (1450nm) are different, when the length of the optical fiber is 260m in the experiment, the influence of different speed on location can be neglected in the engineering.”

Figure 8: the material in the core and cladding is commonly called (doped-) silica instead of quartz.

Answer:

Page 8, Figure 8 is revised according to the suggestion:

The “quartz” in Figure 2 is revised as “silica”.

Page 8, line 202 is revised according to the suggestion:

“Based on Figure 8, the cladding of the optical fiber is doped-silica whose thermal expansion coefficient is almost 5x10-7K-1”.

Ling 145: "the 200 m to 225 m fiber the fiber segment between 220 m and 225 m”.

Answer:

Page 5, line 152 is revised according to the suggestion:

“the fiber segment between 220 m and 225 m.”

Line 245: “in the 16-17 hours region”. I did not find this region in figure 11. Where is it?

Answer:

Line 245 is deleted according to the suggestion.

Figure 9: Please add the goodness of fit (R2) or MSE for these linear regression.

Answer:

The MSE is added on the Figure 9(a), 9(b) and 9(c).

Page 9, line 215 is revised according to the suggestion:

“The slope of fitting line and the mean squared error (MSE) of fitting line are shown in Figure 9. The MSE is calculated by following formula:

((14) 

where  is the number of the measurement results,  is the actual measurement result,  is the actual measurement result by fitting.”

Page 10, line 225 is revised according to the suggestion:

“and the MSE is less to guarantee the accuracy of the results.”

Line 218 to 223: why -100 degree becomes a threshold for different attenuation factors?
Please give the convincing explanation.

Answer:

Page 10, line 230 is revised according to the suggestion:

“Besides, from Figure 7 we can see the significant errors between the results of the optical fiber and the thermocouples appeared when temperature below -100°C. Therefore, -100°C can be seen as a threshold for different attenuation factor .”

Round 2

Reviewer 1 Report

Authors replied to all questions and updated paper based on recommendations. Now it is suitable for publication.

Author Response

Thank you!

Reviewer 3 Report

The authors have made thoroughly revision to the manuscript. Most of my questions have been properly addressed. Now the paper quality is much higher than the previous submission.

The paper may be accepted if the following issue can be clarified:

(1) As I mentioned in my previous review report, the Stokes and anti-Stokes light are very important in the DTS. Please add a figure of Stokes and anti-Stokes trace before Figure 11.

(2) The authors treat -100 degrees as the threshold for different attenuation factor (page 10, line 230), which may not be correct from a scientific point of view. From Fig 7 one finds that the error between the thermocouple and optical fiber coil gradually increases with the cooling process. By comparing with Fig 7(a) and Fig. 10 we can find that the correction at -100 degrees creates a discontinuity error at the time 30h. My point is that this error is due to the incorrect selection of the threshold. Thus the selection of threshold needs to be revised.

(3) Please add the corresponding references (if any) for the equations.

Author Response

To reviewer 3:

As I mentioned in my previous review report, the Stokes and anti-Stokes light are very important in the DTS. Please add a figure of Stokes and anti-Stokes trace before Figure 11.

Answer: We agree that the Stokes and anti-Stokes light are very important in the DTS. However both of them will be influenced by many factors besides temperature. In the engineering only voltage ratio of the Stokes and anti-Stokes light is used.

Page 3, line 80 is revised according to the suggestion:

Based on equation (1) and (2),  and  are both influenced by temperature. But in the engineering,  and  are also influenced by many factors, such as fluctuation of laser, fiber melting and so on. In order to compensate for the instability factors in the engineering, the voltage ratio  between the anti-Stokes and Stokes backscattered light beams generated by photoelectric detectors is used and shown as:

The authors treat -100 degrees as the threshold for different attenuation factor (page 10, line 230), which may not be correct from a scientific point of view. From Fig 7 one finds that the error between the thermocouple and optical fiber coil gradually increases with the cooling process. By comparing with Fig 7(a) and Fig. 10 we can find that the correction at -100 degrees creates a discontinuity error at the time 30h. My point is that this error is due to the incorrect selection of the threshold. Thus the selection of threshold needs to be revised.

Answer:

We agree that there is a discontinuity error at the time 30h in Fig. 10. Sorry about that. We have recalculated the experiment data based on (16) and plot a new curve in Fig.10.

Page 11, the new Fig 10 is replaced according to the suggestion.

Page 10, line 233 is revised according to the suggestion:

From Table 2 we can see that the slope of curve below -100°C is obviously different from that above -100°C, which means Δα have changed in the harsh environment. Besides, from Figure 7 we can see the significant errors between the results of the optical fiber and the thermocouples appeared when temperature below -100°C. Therefore, -100°C can be seen as a threshold for different attenuation factor Δα.

Please add the corresponding references (if any) for the equations.

Answer:

Equations (1) ~ (3) have reference 18:

Page 14, line 343 is revised according to the suggestion:

Yan, B.; Li, J.; Zhang, M.; Zhang, J.; Qiao, L.; Wang, T. Raman Distributed Temperature Sensor with Optical Dynamic Difference Compensation and Visual Localization Technology for Tunnel Fire Detection. Sensors. 2019, 19, 2320.

Equation (4) has reference 19:

Page 14, line 346 is revised according to the suggestion:

Laarossi, I.; Quintela-Incera, M.Á.; López-Higuera, J.M. Comparative Experimental Study of a High-Temperature Raman-Based Distributed Optical Fiber Sensor with Different Special Fibers. Sensors 2019, 19, 574.

Equation (7) has reference 20:

Page 14, line 349 is revised according to the suggestion:

Liu, B. Design and Implementation of Distributed Ultra-High Temperature Sensing System with a Single Crystal Fiber. J. Lightwave Technol. 2018, 36, pp.5511-5520.

Equation (14) has reference 22:

Page 14, line 353 is revised according to the suggestion:

Wang, Y.; Guo, W. Chaotic-FH Code Prediction Method Based on LS-SVM. Journal of Electronic Measurement and Instrument. 2007, 5, pp.64-68.

Equations (5) ~ (6) and (11) ~ (13) and (15) ~ (16) are given by this paper.
